# Novel Agents as Main Drivers for Continued Improvement in Survival in Multiple Myeloma

**DOI:** 10.3390/cancers15051558

**Published:** 2023-03-02

**Authors:** Borja Puertas, Verónica González-Calle, Eduardo Sobejano-Fuertes, Fernando Escalante, José A. Queizán, Abelardo Bárez, Jorge Labrador, José María Alonso-Alonso, Alfonso García de Coca, Alberto Cantalapiedra, Teresa Villaescusa, Carlos Aguilar-Franco, Elena Alejo-Alonso, Beatriz Rey-Bua, Lucía López-Corral, Ramón García-Sanz, Noemi Puig, Norma C. Gutiérrez, María-Victoria Mateos

**Affiliations:** 1Instituto de Investigación Biomédica de Salamanca (IBSAL), Cancer Research Center-IBMCC (USAL-CSIC), CIBERONC, University Hospital of Salamanca, 37007 Salamanca, Spain; 2Department of Hematology, University Hospital Dr. José Molina Orosa (Lanzarote, Canary Islands), 35500 Las Palmas, Spain; 3Department of Hematology, University Hospital of León, 24071 León, Spain; 4Department of Hematology, University Hospital of Segovia, 40002 Segovia, Spain; 5Department of Hematology, University Hospital of Ávila, 05004 Ávila, Spain; 6Department of Hematology, University Hospital of Burgos, 09006 Burgos, Spain; 7Department of Hematology, University Hospital of Rio Carrión (Palencia), 34005 Palencia, Spain; 8Department of Hematology, University Clinical Hospital of Valladolid, 47003 Valladolid, Spain; 9Department of Hematology, University Hospital of Rio Hortega (Valladolid), 47012 Valladolid, Spain; 10Department of Hematology, University Hospital of Virgen de la Concha (Zamora), 49022 Zamora, Spain; 11Department of Hematology, University Hospital of Soria, 42005 Soria, Spain

**Keywords:** novel agents, survival, long survivors

## Abstract

**Simple Summary:**

Survival of patients with multiple myeloma continues to improve over time in parallel with the development of new treatments. The combination of novel agents in the first-line setting appears to be the main factor driving improvement, resulting in overall survival (OS) of more than 12 and 8 years in patients aged ≤ 70 and >70 years, respectively. In addition, different baseline characteristics have been identified that favorably affected the probability of long-term survival (≥10 years) compared with early death (≤2 years). Therefore, we could conclude that multiple myeloma has become a chronic and even curable disease, in a subset of patients with the current therapeutic approaches.

**Abstract:**

(1) Background: New therapeutic strategies have improved the prognosis of multiple myeloma (MM), changing the accepted view of this disease from being incurable to treatable. (2) Methods: We studied 1001 patients with MM between 1980 and 2020, grouping patients into ten-year periods by diagnosis 1980–1990, 1991–2000, 2001–2010 and 2011–2020. (3) Results: After 65.1 months of follow-up, the median OS of the cohort was 60.3 months, and OS increased significantly over time: 22.4 months in 1980–1990, 37.4 months in 1991–2000, 61.8 months in 2001–2010 and 103.6 months in 2011–2020 (*p* < 0.001). Using novel agents in the front-line setting for myeloma patients yielded a significantly better OS than in those treated with conventional therapies, especially when combinations of at least two novel agents were used. The median OS of patients treated with the combination of at least two novel agents in induction was significantly prolonged compared to those treated with a single novel agent or conventional therapy in induction: 143.3 vs. 61.0 vs. 42.2 months (*p* < 0.001). The improvement was apparent in all patients regardless of age at diagnosis. In addition, 132 (13.2%) patients were long-term survivors (median OS ≥ 10 years). Some independent clinical predictors of long-term survival were identified: ECOG < 1, age at diagnosis ≤ 65 years, non-IgA subtype, ISS-1 and standard-risk cytogenetic. Achieving CR and undergoing ASCT were positively associated with >10 years of survival. (4) Conclusions: The combination of novel agents appears to be the main factor for the improvement in survival in MM, which is becoming a chronic and even curable disease in a subtype of patients without high-risk features.

## 1. Introduction

A better understanding of the biology of multiple myeloma (MM), the development of research and the introduction of new therapeutic strategies have improved the outcome of patients with MM and changed the dogma of MM from being an incurable to being a treatable or even curable disease [1].

The combination of melphalan and prednisone (MP) was introduced in the 1960s, after which autologous stem cell transplantation (ASCT) has been the only therapeutic change in the field of myeloma until the beginning of the 21st century [2,3]. We have witnessed major advances in myeloma treatment in the last twenty years, such that the introduction of novel agents, such as proteasome inhibitors (PIs) and immunomodulators (IMiDs), is regarded as the foremost cause of improved survival in myeloma patients of all age groups, but of young patients in particular [4,5,6,7,8,9,10]. In addition, the incorporation of anti-CD38 antibodies, first in the relapse and later in the upfront settings, has prompted a paradigm shift, notably for transplant-ineligible patients. Currently, the standard of care (SoC) for MM consists of a combination of PIs and/or IMiDs and/or anti-CD38 antibodies [11,12], followed by ASCT and maintenance when the patient is eligible [13,14,15,16,17]. Although several studies have reported the benefit to survival over time, none of them has focused on how the impact of first-line therapy on real-world outcomes has changed over the years. 

Various prognostic factors have been identified that allow us to distinguish high-risk patients and patients with a very durable response. In this regard, the International Staging System (ISS) [18] and the chromosomal aberrations detected with FISH (fluorescence in situ hybridization) [19] are the main prognostic factors used to categorize MM patients. In addition, several studies have identified an association between clinical and biological characteristics, depth of response and increased survival [20,21,22,23]. These prognostic advances, along with the introduction of novel therapies, have reignited interest in whether a cure is possible in a subset of myeloma patients. 

Based on this background, a retrospective study of patients diagnosed and treated over 40 years was undertaken to evaluate whether the outcomes for myeloma patients have improved over this period, the potential role of novel agents introduced over time and the clinical and biological characteristics that may predict long-term survival.

## 2. Materials and Methods

This retrospective observational study was designed to include patients correlatively diagnosed with MM and those who underwent an ASCT at the University Hospital of Salamanca between 1980 and 2020. Patients with smoldering myeloma or plasma cell leukemia were not included. The follow-up cut-off date was 31 October 2022. The ethical committee of the University Hospital of Salamanca approved the study, which was conducted in accordance with the 1964 Declaration of Helsinki.

Patients were divided into four groups according to the decade in which they were diagnosed. These periods encompass distinct approaches to treatment. During the first period (1980–1990), chemotherapy, especially MP, was the SoC for most patients; in the second period (1991–2000), ASCT had become the SoC for transplant-eligible patients; the third period (2001–2010) was characterized by the increasing prevalence of treatment with PI and IMIDs; and the most recent period (2011–2020) featured the introduction of anti-CD38 monoclonal antibodies. Patients were also divided into two groups according to age at diagnosis: ≤70 years and >70 years because this is the cut-off age considered for eligibility for ASCT in our clinical practice.

Novel agent-based inductions include those containing PIs, IMIDs and anti-CD38 monoclonal antibodies; conventional therapy-based inductions are referred to as conventional chemotherapy. 

The effect of novel agents was analyzed according to their number included in the induction therapy: inductions with a single novel agent (e.g., bortezomib, melphalan and prednisone (VMP), lenalidomide and dexamethasone (Rd), etc.) and inductions with at least two novel agents (e.g., bortezomib, thalidomide and dexamethasone (VTD), bortezomib, lenalidomide and dexamethasone (VRD), daratumumab plus VMP, etc.).

The cytogenetic status of patients was assessed with FISH, as previously reported, in non-separated plasma cells until 2005 [24] and in separated plasma cells thereafter [25]. Illegitimate translocations of the *IGH* gene, t (11;14), t (4;14) and t (14;16), deletion 17p (del17p) and chromosome 1 abnormalities (1q gain and 1p deletion since 2010) were analyzed. A threshold of 10% was used as a cut-off for translocations and 20% for numerical aberrations, according to the European Myeloma Network [26]. Patients were defined as high-risk based on the International Myeloma Working Group (IMWG) [27]. 

The response was evaluated according to the 2016 IMWG criteria [28]. The overall rate response (ORR) was defined as the percentage of patients who achieved partial response (PR) or better. A single complete response (CR) category was established that pooled complete and stringent complete responses. 

OS was defined as the time from diagnosis until the date of death or last follow-up. Long-term survivors were defined as patients who had lived for at least 10 years following their diagnosis of MM. Early death was defined as a death occurring within 2 years of diagnosis, from whatever cause.

Chi-square tests identified statistically significant differences between the proportions of categories of qualitative variables, including the ORR and percentage of CR, and the associated odds ratio (OR) and 95% confidence interval (CI) were estimated using logistic regression. An ANOVA test was used to compare the median of quantitative variables. The differences in OS were defined using the log-rank test, and the corresponding hazard ratio (HR) and 95% CI were estimated using Cox regression. Univariable and multivariable logistic regression was used to compare long-term survivors and early-death patients. Values of *p* < 0.05 were considered statistically significant. Statistical analyses were performed with IBM SPSS Statistics version 26.

## 3. Results

A total of 1001 patients were diagnosed between 1980 and 2020: 93 (9.3%) during 1980–1990, 178 (17.8%) during 1991–2000, 314 (31.4%) during 2001–2010 and 416 (41.5%) in the most recent period (2011–2020). The median age at diagnosis was 64 years (range, 28–93 years); 297 (31.0%) patients were diagnosed at more than 70 years of age, and 567 (56.6%) were men. The median follow-up period of the entire cohort was 65.1 months (range, 2.4–382.7 months), and the median number of the lines of treatment was 1 (1–14). Four-hundred and ninety-eight patients (49.8%) underwent ASCT. Table 1 shows the baseline characteristics of patients by diagnostic period. Patients diagnosed in the early decades more frequently presented anemia, hypercalcemia, renal failure and a poorer ECOG (European Cooperative Oncology Group) PS (performance status) than those diagnosed more recently.

The median OS of the entire cohort was 60.3 months (95% CI, 54.2–62.4 months). A progressive increase in OS was observed over time (Figure 1). The median OS improved from 22.4 months in 1980–1990 (HR 3.5 [95% CI, 2.5–4.4]; *p* < 0.001), to 37.4 months in 1991–2000 (HR 2.1 [95% CI, 1.7–2.7]; *p* < 0.001), to 61.8 months in 2001–2010 (HR 1.4 [95% CI, 1.2–1.8]; *p* = 0.001) and to 103.6 months in 2011–2020 (reference decade).

Patients diagnosed when they were ≤70 years of age tended to have a longer OS than those diagnosed at >70 years (93.0 vs. 28.2 months; HR 3.0 [95% CI, 2.5–3.5]; *p* < 0.001). OS improved in both groups over time. Young patients (≤70 years) diagnosed in the last decade (2011–2020) had a longer median OS (127.6 months) than those diagnosed in the other periods: 39.9 months during 1980–1990 (HR 4.2 [95% CI, 2.8–6.2]; *p* < 0.001); 53.5 months during 1991–2000 (HR 2.5 [95% CI, 1.8–3.5]; *p* < 0.001); and 96.6 months during 2001–2010 (HR 1.6 [95% CI, 1.2–2.2]; *p* = 0.002) (Figure 2A). In addition, a modest but significant benefit was found in elderly-diagnosed patients (>70 years). The median OS of elderly patients diagnosed between 2011 and 2020 (32.4 months) was longer than that of patients diagnosed in the other decades: 18.8 months during 1980–1990 (HR 1.9 [95% CI, 1.2–3.2]; *p* = 0.007) and 16.4 months during 1991–2000 (HR 1.8 [95% CI, 1.2–2.7]; *p* = 0.002). Furthermore, the median OS of patients diagnosed in 2011–2020 tended to be longer than that of those diagnosed during 2001–2010 (28.3 months; HR 1.3 [95% CI, 1.0–1.8]; *p* = 0.060) (Figure 2B).

### 3.1. Impact of the Introduction of Novel Agents on Outcomes

Having observed a significantly increasing benefit to the survival of MM patients diagnosed over time, we analyzed the effect on survival of receiving novel agents in the upfront setting over the forty-year period. Considering the entire cohort, 503 patients (50.2%) were treated with novel agents in the first line, 480 (48.0%) received conventional therapies, and 18 (1.8%) did not receive any treatment due to poor performance status. Of the patients who received novel agents, 230 (45.8%) were treated with single novel agent inductions and 272 (54.2%) received a combination of at least two novel agents in induction. The first line of treatment of the entire cohort, by age and decade of diagnosis, is summarized in Table 2. Overall, there was an increase in the use of novel agent-based inductions over time. Ninety percent of patients diagnosed in the most recent decade were treated with new drugs in the first line. Notably, the combination of at least two novel agents in induction was more frequent in the 2011–2020 period than in the previous decade (67.5% vs. 13.7%; *p* < 0.001). ASCT has been progressively more often incorporated into clinical practice for the treatment of younger patients over time. Only 10.2% of patients with a diagnosis at age ≤ 70 years underwent ASCT during the first period (1980–1990) compared with 59.6% during 1991–2000 and 83.4% of patients in the last twenty years (*p* < 0.001).

In terms of response, PR or better was more likely to be achieved through the use of novel agents in induction than with conventional therapies (91.3% vs. 73.0%, OR 3.9 [95% CI, 2.7–5.7]; *p* < 0.001). In addition, the probability of achieving CR after induction was significantly higher in patients treated with novel agents (38.3% vs. 17.7%; OR 2.9 [95% CI, 2.1–3.9]; *p* < 0.001). Patients treated with the combination of at least two novel agents in induction responded better than those who received single novel agent inductions, with respect to achieving PR or better (97.4% vs. 83.8%; OR 7.2 [95% CI, 3.1–16.5]; *p* < 0.001) and CR (46.8% vs. 27.8%; OR 2.3 [95% CI, 1.6–3.4]; *p* < 0.001).

This improved response led to an increase in OS in MM patients treated with novel agents in the first line: 107.6 vs. 42.2 months (HR 2.1 [95% CI, 1.7–2.4]; *p* < 0.001). However, it is worth noting that the improvement observed in patients treated with novel agents arose mainly from treatments with at least two novel new drugs in the first line. The median OS of those treated with the combination of at least two novel agents in induction was significantly longer than that of patients who received single novel agent inductions: 143.3 vs. 61.0 months (HR 2.2 [95% CI, 1.6–2.9]; *p* < 0.001) (Figure 3).

Furthermore, we analyzed the effect of novel agents on the response (Appendix A) and survival of young and elderly patients, to overcome the transplant-eligibility and comorbidity biases. 

In the ≤70-year group, the likelihood of achieving PR or better and CR after induction was greater in patients treated with novel agent-based inductions compared with those receiving conventional therapies (92.8% vs. 80.3%; OR 3.1 [95% CI, 1.9–5.2]; *p* < 0.001; and 39.6% vs. 20.1%; OR 2.6 [95% CI, 1.8–3.7]; *p* < 0.001). Both responses were significantly stronger in patients treated with the combination of at least two novel agents in induction compared with those who received single novel agent inductions (PR or better: 96.9% vs. 85.7%, OR 5.2 [95% CI, 2.1–12.8], *p* < 0.001; and CR: 46.0% vs. 28.6%, OR 2.1 [95% CI, 1.3–3.4], *p* = 0.001). In addition, the median OS was significantly longer in young patients treated with novel agents in induction than those who received conventional therapies: 143.3 vs. 65.7 months (HR 2.0 [95% CI, 1.6–2.5]; *p* < 0.001). The combination of at least two novel agents in induction led to a longer OS (143.3 months) than in those treated with single novel agent inductions (113.0 months; HR 1.8 [95% CI, 1.2–2.6]; *p* = 0.004) (Figure 4A). 

Considering patients older than 70 years at diagnosis, those who received novel agents had better responses than those receiving conventional therapies, achieving at least PR (87.1% vs. 68.0%; OR 3.2 [95 CI%, 1.7–6.1]; *p* < 0.001) and CR after induction (34.7% vs. 16.8%; OR 2.6 [95% CI, 1.4–4.8]; *p* = 0.002). Both responses were better among the patients treated with a combination of at least two novel agents in induction than among patients treated with single novel agent inductions (PR or better: 100.0% vs. 80.7%, OR not estimated, *p* = 0.001; and CR: 51.2% vs. 26.5%, OR 2.9 [95% CI, 1.3–6.4], *p* = 0.007). The introduction of novel agents boosted OS with respect to the outcome of patients treated with conventional therapies: 49.5 vs. 26.8 months (HR 1.9 [95% CI, 1.4–2.5]; *p* < 0.001). In addition, patients treated with a combination of at least two novel agents in induction had a significantly longer OS than those receiving single novel agent inductions: 101.8 vs. 43.7 months (HR 1.7 [95% CI, 1.1–2.8]; *p* = 0.034) (Figure 4B). 

### 3.2. Long-Term Survivors

Overall, 132 (13.2%) patients were considered long-term survivors (median OS ≥ 10 years) in our entire cohort. Five patients (3.8%) were diagnosed between 1980 and 1990, twenty-five (18.9%) during 1991–2000, seventy-six (57.6%) during 2001–2010 and twenty-six (19.7%) in the most recent decade (2011–2020). The median age at diagnosis was 57 years (range, 29–79 years) and 66 (50.0%) were men. The median number of treatment lines was 2 (range, 1–14) and 105 (79.5%) underwent ASCT (OR 4.6 [95% CI, 2.9–7.1]; *p* <≈ 0.001). In addition, achieving CR or better after the first line of treatment was positively associated with living at least 10 years following an MM diagnosis (19.0% vs. 10.6%; OR 2.0 [95% CI, 1.4–2.9]; *p* < 0.001). To identify clinical predictors of long-term survival, we compared baseline clinical characteristics at the time of diagnosis of long-term survivors with those of patients who died early (OS ≤ 2 years, 252 patients (25.2%) in the complete cohort). In the univariable model (Table 3), age at diagnosis ≤ 65 years, IgG and non-IgA subtypes, bone marrow plasma cell infiltration < 30%, ECOG PS 0-1, hemoglobin levels ≥ 10 g/dL, creatinine levels ≤ 2 mg/dL, calcium levels < 11 mg/dL, albumin levels ≥ 3.5 g/dL, β_2_ microglobulin levels < 3.5 and 5 mg/dL, standard-risk cytogenetics, being ISS-1 and not being ISS-3 were characteristics associated with over 10-year survival. In the multivariable model (Table 4), age at diagnosis ≤65 years, the non-IgA subtype, ECOG PS 0-1, standard risk cytogenetics and being ISS-1 were characteristics associated with a 10-year survival. 

## 4. Discussion

This retrospective study confirms the continuing improvement in the survival of myeloma patients over time. The combination of new drugs in the first-line setting appears to be the main factor driving improvement in the most recent period. Moreover, to our knowledge, this is the first historical study to report the effect on outcomes of the combination of PIs and IMiDs and the positioning of anti-CD38-based inductions in first-line therapy. In this regard, patients in our cohort treated with the combination of at least two novel agents in induction showed an encouraging median OS of more than 10 years. In addition, age at diagnosis ≤65 years, non-IgA subtype, ECOG PS 0-1, standard risk cytogenetics and being ISS-1 were identified as favorable prognostic factors associated with long-term survival (≥10 years). 

Many retrospective and population-based investigations have reported that the improvement in survival of patients with MM over time was due to the introduction of ASCT and novel agents. Notably, the increase in OS was observed in younger-diagnosed patients (≤65 years) [4,5,6,9,29,30,31,32]. Kumar and colleagues reported the OS of almost 3000 patients diagnosed at the Mayo Clinic, dividing patients into two groups according to the year of diagnosis: 1971–1996 and 1997–2006. Sixty percent of patients diagnosed in the most recent period received novel agents in the first line, and this group presented better survival (44.8 vs. 22.9 months; *p* < 0.001) but was limited to patients aged ≤ 65 years (60.0 vs 33.0 months; *p* < 0.05) [4]. Consistent results were also reported by the Greek group of myeloma. In this study, patients diagnosed between 1985 and 1999 were compared with those diagnosed in ≥2000, in which only 30% received novel agents. The median OS of patients diagnosed in 2000 or after was significantly longer (44,8 vs. 22.9 months; *p* < 0.001). This benefit in OS was exclusively shown in patients ≤ 65 years (not reached vs. 42,0 months; *p* < 0.001) [5]. The lack of benefit of novel agents in older patients was limited by their underuse in this population. However, various studies have shown the benefit of using novel agents in the more elderly population [7,8,10,33,34]. Kumar et al. first reported improved survival in patients older than 65 years due to the novel agents [8]. In this retrospective study at the Mayo Clinic, 1038 patients diagnosed between 2001 and 2010 were analyzed, grouping patients into two 5-year periods by diagnosis (2001–2005 and 2006–2010). Sixty percent of the cohort was treated with novel agents in the first line. Patients diagnosed in the most recent period achieved significantly prolonged OS (6.1 vs. 4,6 years; *p =* 0.002), including patients aged >65 years (5.0 vs. 3.2 years; *p* < 0.05). No differences were observed in younger patients because of the effect of ASCT and probably the lack of follow-up. A historic study on the Hospital Clinic of Barcelona showed a significant improvement in OS over decades, regardless of the age at diagnosis, but especially in younger patients [35]. In addition, the population-based studies indirectly support the benefit of different milestones in MM treatment. Although these studies did not include clinical and treatment data, the survival of patients improved over time, indicating the positive impact of the new approach on myeloma patients [10,30,31,32,33,34,36,37]. Indeed, this progressive improvement in survival was also observed in our cohort. Figure 1 illustrates how outcomes have improved decade upon decade, reflecting the therapeutic advances in myeloma treatment. Thus, the median OS of the second calendar period (1991–2000) is longer than that of the first decade (1980–1990) due to the introduction of ASCT (59.6% vs. 10.2%); patients diagnosed in the third period (2001–2010) generally had better OS because of the introduction of novel agents; and finally, in the most recent decade (2011–2020), the combination of novel agents in induction led to longer survival than patients diagnosed in the third decade. The improved results over time, especially since 2000, were replicated when we stratified the population by age at diagnosis (Figure 2A,B). This is a reflection of the increased use of novel agent-based inductions among younger and older patients, and future approaches will require frail-adapted therapy for the elderly to be planned in order to obtain population outcomes comparable with those of the younger population.

As might be expected, patients who received novel-agent inductions presented twice the median OS of those treated with chemotherapy or polychemotherapy in the younger and older groups. Therefore, our results are in line with those reported by other authors, whereby the new agents have improved the outcomes of myeloma patients, regardless of age at diagnosis. However, the number of novel agents is important: a notable finding from our analysis is that the increasingly widespread use of combinations with at least two novel agents in induction is one of the main factors that has improved survival in recent years. In this regard, almost 100% and 50% of patients treated with the combination of at least two novel agents in induction achieved at least PR and CR, respectively, and presented excellent OS (Figure 3). Thus, patients aged ≤ 70 years and >70 years treated using this approach presented a striking median OS of 12 and 8 years, respectively (Figure 4A,B). These results are consistent with the phase 3 trials conducted in transplant-eligible [13,14,15,16,17] and ineligible patients [11,12]. However, considering the older group, the OS in the last decade (32.4 months) was lower than expected. Although some of our elderly patients participated in the aforementioned trials, most of them had not been selected. Therefore, for the sample of elderly patients treated with a combination of at least two novel agents (*n* = 39), their heterogeneity, comorbidities and age-related frailty were the main reasons for the poorer survival. Notably, increased mortality was observed in patients aged > 70 years in the first 24 months, regardless of the induction therapy. Establishing frailty scales in daily practice is necessary if each patient is to be offered the most appropriate treatment [38,39], given that the progressive introduction of novel drugs seems to improve their outcomes. In addition, the snowball approach would be a useful strategy in frail patients. This strategy consists of adding drugs progressively as the patient’s performance status improves until the optimal treatment is achieved. An ongoing phase 3, multicenter, randomized trial by the Spanish Myeloma Group, the GEM2017FIT trial, aims to determine the optimal treatment for newly diagnosed elderly MM patients aged between 65 and 80 years. Another study (IBERDARADEX) is testing iberdomide-based combinations in elderly patients, although 30% of the different cohorts are frail patients to enable safety and efficacy to be evaluated in this specific subgroup of patients.

A further noteworthy finding is the progressive increase over the 40 years of the study in the proportion of patients considered to be long-term survivors. There was a five-fold increase in the number of patients living 10 years from 1980–1990 to 1991–2000 and a three-fold increase from 1991–2000 to 2001–2010. The number of long-term survivors in the decade 2011–2020 could not be reliably evaluated due to the intrinsically insufficient follow-up (51.4 months). However, despite this limitation, the same number of patients have already survived more than 10 years in the last decade as in 1991–2000. This increase in the number of long-term survivors is mainly due to the introduction of new therapies, both in induction and relapse. Many of these patients have received second-generation PIs and IMiDs, anti-CD38 monoclonal antibodies and targeted therapies in the relapsed/refractory setting. Moreover, eight out of ten patients underwent ASCT, which was associated with an almost five-fold greater probability of achieving prolonged survival. Two studies have reported that 20–30% of transplanted patients achieved an OS longer than 10 years [40,41]. One in five patients who achieved CR or better after the first line of treatment were long-term survivors. Some authors have already directly associated the achievement of CR with longer survival in transplant-eligible [42] and ineligible patients [21,22]. This finding is very encouraging given that the introduction of new drugs in the first line has been shown to increase the likelihood of achieving CR, especially if patients receive the combination of at least two novel agents in induction. In addition, age at diagnosis ≤ 65 years, non-IgA subtype, ECOG PS 0-1, standard risk cytogenetics and ISS-1 favorably and independently affected the probability of long-term survival (≥10 years) compared with early death (≤2 years). The main baseline characteristics associated with the tumor burden (hemoglobin, calcium, renal failure, plasmacytosis, etc.) have been reported as independent prognostic factors of long-term survival [20], but the loss of diagnostic data could have influenced the estimates of the multivariable model.

The foremost limitation of the present study is its retrospective nature, especially with respect to the lack of some clinical or laboratory data, and the intrinsically short follow-up of patients diagnosed in the most recent decade. The cohort studied is younger than expected, probably due to the fact that it is a transplant referral center. No distinction was made between fixed-duration and continuous treatments in the survival analysis. In addition, no specific survival sub-analysis of anti-CD38 was performed. However, many elderly patients treated with the combination of at least two novel agents in induction received anti-CD38-based regimens. The strength of this study lies in it involving a large cohort with a long follow-up that provides real-world evidence and shows the evolution of myeloma patients over time. Nevertheless, our findings should be confirmed in independent studies and with further follow-up. 

## 5. Conclusions

In summary, this study shows the continuing increase in survival in MM as time goes by, regardless of the age at diagnosis, although improvements are particularly marked in the younger population. The most recent improvements in the response and survival of myeloma patients are due to the introduction of novel agents in the front-line setting, especially when at least two novel agents are combined in induction. This finding emphasizes the importance of using the most effective combinations in first-line therapy, as previously reported [43]. The likelihood of being a long-term MM survivor is increasing because of the use of more, better and less toxic therapies in the first line and relapse. Achieving CR and undergoing ASCT were positively associated with survival longer than 10 years. Patients with good performance status diagnosed with ≤65 years of non-IgA myeloma, ISS-1 and without high-risk cytogenetics had excellent survival. These findings lead us to conclude that myeloma has become a chronic disease, and even a curable one, in a subset of patients. To achieve this goal, we have begun to incorporate risk and response-adapted therapy into our clinical practice, mainly through the Spanish Myeloma Group’s clinical research. This allows us to stratify the elderly population according to their frailty and to treat standard and high-risk patients in different ways, as appropriate, or even to intensify or de-escalate treatment based on the presence of measurable residual disease detected using highly sensitive techniques for assessing response. These new factors will be incorporated into the analyses of our forthcoming studies.

## Figures and Tables

**Figure 1 cancers-15-01558-f001:**
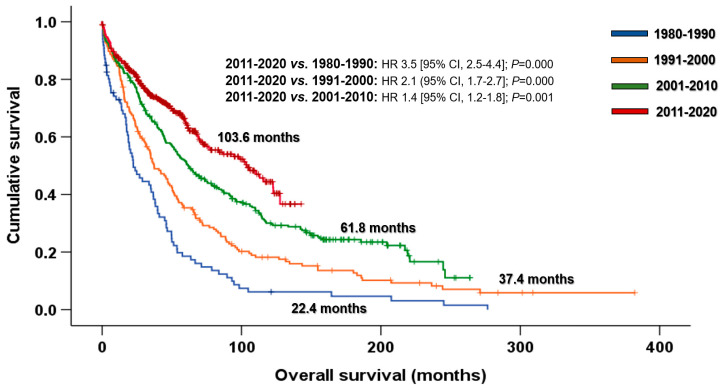
Overall survival by decades. Abbreviations: CI: confidence interval; HR: hazard ratio.

**Figure 2 cancers-15-01558-f002:**
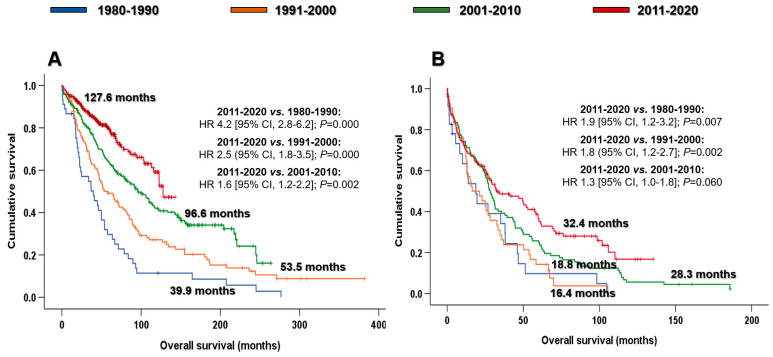
Overall survival by decades and age at diagnosis: (**A**) ≤70 years and (**B**) >70 years. Abbreviations: CI: confidence interval; HR: hazard ratio.

**Figure 3 cancers-15-01558-f003:**
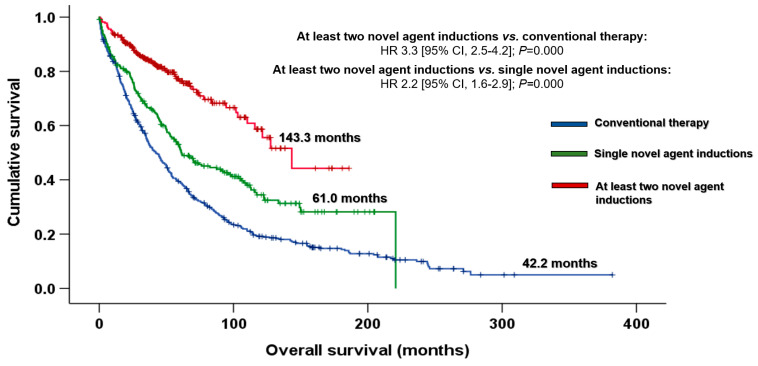
Overall survival by induction therapy. Abbreviations: CI: confidence interval; HR: hazard ratio.

**Figure 4 cancers-15-01558-f004:**
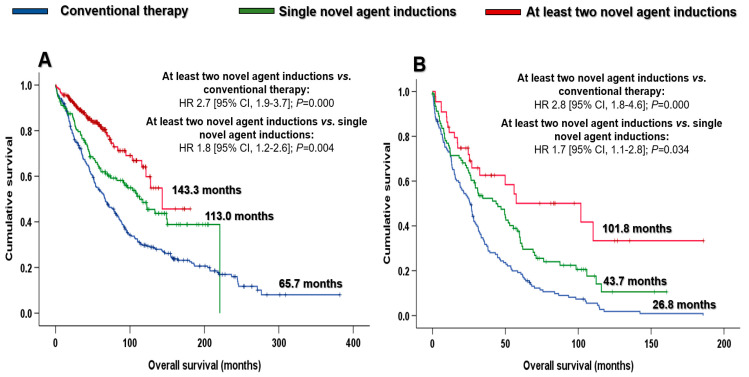
Overall survival by induction therapy and age at diagnosis: (**A**) ≤70 years and (**B**) >70 years. Abbreviations: CI: confidence interval; HR: hazard ratio.

**Table 1 cancers-15-01558-t001:** Baseline characteristics of the entire cohort.

	All Patients (*n* = 1001)	Group 1 (1980–1990) (*n* = 93)	Group 2 (1991–2000) (*n* = 178)	Group 3 (2001–2010) (*n* = 314)	Group 4 (2011–2020) (*n* = 416)	*p* Value
Follow-up in months, median (range)	65.1 (2.4–382.1)	121.3 (3.0–121.7)	119.0 (15.0–382.1)	157.8 (10.3–263.8)	51.4 (2.4–142.9)	
Age at diagnosis, median (range) ^a^	64 (28–93)	68 (38–86)	64 (31–88)	63 (28–89)	65 (30–93)	0.153
Age at diagnosis ≤70, *n* (%) Older than 70, *n* (%)	662 (69.0) 297 (31.0)	49 (66.2) 25 (26.9)	114 (72.6) 43 (27.4)	215 (68.9) 97 (31.1)	284 (68.3) 132 (31.7)	0.722
Gender, male, *n* (%)	567 (56.6)	44 (47.3)	96 (53.9)	173 (55.1)	254 (61.1)	0.059
Ig isotype, *n* (%) ^b^ IgG IgA IgM IgD Light chains only Non-secretory	557 (56.6) 252 (25.6) 2 (0.2) 7 (0.7) 145 (14.7) 21 (2.1)	45 (49.5) 33 (36.3) 0 (0.0) 0 (0.0) 13 (14.3) 0 (0.0)	87 (49.7) 54 (30.9) 0 (0.0) 7 (4.0) 27 (15.4) 0 (0.0)	187 (59.9) 72 (23.1) 1 (0.3) 0 (0.0) 37 (11.9) 15 (4.8)	238 (58.6) 93 (22.9) 1 (0.2) 0 (0.0) 68 (16.7) 6 (1.5)	0.061 0.014 - - 0.328 -
Light chain isotype Kappa, *n* (%) ^c^	575 (59.1)	39 (46.4)	100 (57.8)	187 (60.9)	249 (60.9)	0.026
ECOG PS 0–1, *n* (%) ^d^	497 (64.0)	27 (34.6)	53 (44.5)	132 (64.7)	285 (75.8)	0.000
Hb ≤ 10 g/dL, *n* (%) ^e^	366 (39.6)	51 (56.0)	75 (46.9)	113 (38.6)	127 (33.3)	0.000
Cr ≥ 2 mg/dL, *n* (%) ^f^	185 (19.6)	25 (27.8)	33 (20.4)	50 (16.9)	77 (19.4)	0.157
Ca ≥ 11 mg/dL, *n* (%) ^g^	144 (16.2)	27 (30.7)	25 (16.4)	36 (13.2)	56 (14.8)	0.001
Lytic lesions, *n* (%) ^h^	617 (69.2)	68 (76.4)	106 (66.3)	164 (59.9)	279 (75.8)	0.000
Elevated LDH, *n* (%) ^i^	230 (43.5)	No data	24 (72.7)	136 (72.0)	70 (22.8)	0.000
Albumin, g/dL, mean (SD)	3.6 (±0.7)	3.6 (±0.7)	3.7 (±0.7)	3.5 (±0.7)	3.6 (±0.7)	0.476
β2 microglobulin, mg/dL, mean (SD)	5.8 (±5.4)	5.5 (±4.1)	6.6 (±7.6)	4.7 (±4.2)	6.5 (±5.2)	0.000
ISS, *n* (%) ^j^ I II III	280 (33.5) 280 (33.5) 277 (33.0)	15 (38.5) 10 (25.6) 14 (35.9)	62 (42.5) 37 (25.3) 47 (32.2)	94 (34.6) 115 (42.3) 63 (23.2)	109 (28.7) 118 (31.1) 153 (40.3)	0.020 0.001 0.000
High-risk cytogenetic ^k,^*, *n* (%)	116 (18.3)	No data	4 (25.0)	39 (16.0)	73 (19.8)	0.396

Abbreviations: Ig: immunoglobulin; ECOG: European Cooperative Oncology Group; PS: performance status; Hb: hemoglobin; Cr: creatinine; Ca: calcium; LDH: lactate dehydrogenase; ISS: International Staging System; SD: standard deviation. Data were available in 959 (a), 984 (b), 973 (c), 777 (d), 925 (e), 944 (f), 891 (g), 891 (h), 529 (i), 837 (j) and 627 patients (k). * High-risk cytogenetic was considered t(4;14), t(14;16) and del17p.

**Table 2 cancers-15-01558-t002:** Treatments used by age and decade of diagnosis.

	Age ≤ 70 Years (*n* = 662)				Age Older than 70 Years (*n* = 279)			
	1980–1990 (*n* = 49)	1991–2000 (*n* = 114)	2001–2010 (*n* = 215)	2011–2020 (*n* = 284)	1980–1990 (*n* = 21) ^a^	1991–2000 (*n* = 40) ^b^	2001–2010 (*n* = 94) ^c^	2011–2020 (*n* = 124) ^d^
Lines of therapy, median (range)	1 (1–4)	2 (1–7)	2 (1–14)	1 (1–9)	1 (1–2)	1 (1–2)	2 (1–5)	2 (1–8)
Chemotherapy (CyP, MP)	25 (51.0)	15 (13.1)	6 (2.8)	2 (0.7)	13 (61.9)	32 (80.0)	57 (60.6)	18 (14.5)
Polychemotherapy (VBCMP, VBAD, VAD)	24 (49.0)	97 (85.1)	116 (54.0)	9 (3.2)	8 (38.1)	8 (20.0)	4 (4.3)	0 (0.0)
Novel agents in first line	0 (0.0)	0 (0.0)	93 (43.2)	273 (96.1)	0 (0.0)	0 (0.0)	32 (34.0)	105 (84.7)
1 novel agent in first line	0 (0.0)	0 (0.0)	80 (37.2)	57 (20.1)	0 (0.0)	0 (0.0)	27 (28.7)	66 (53.2)
≥2 novel agents in first line	0 (0.0)	0 (0.0)	13 (6.0)	216 (76.1)	0 (0.0)	0 (0.0)	5 (5.3)	39 (31.5)
PI-based scheme (VD, VMP, VCD, PAD…)	0 (0.0)	0 (0.0)	69 (32.1)	54 (19.0)	0 (0.0)	0 (0.0)	22 (23.4)	58 (46.8)
IMID-based scheme (TD, TCD, TAD, Rd…)	0 (0.0)	0 (0.0)	11 (5.2)	3 (1.1)	0 (0.0)	0 (0.0)	5 (5.3)	11 (8.9)
PI plus IMID (VTD, VRD…)	0 (0.0)	0 (0.0)	13 (6.0)	195 (68.7)	0 (0.0)	0 (0.0)	5 (5.3)	10 (8.1)
Anti-CD38-based scheme (Any combination which included anti-CD38)	0 (0.0)	0 (0.0)	0 (0.0)	19 (6.7)	0 (0.0)	0 (0.0)	0 (0.0)	26 (21.0)
Others	0 (0.0)	2 (1.8)	0 (0.0)	2 (0.7)	0 (0.0)	0 (0.0)	1 (1.1)	1 (0.8)
ASCT	5 (10.2)	68 (59.6)	179 (83.3)	237 (83.5)	0 (0.0)	0 (0.0)	0 (0.0)	3 (2.3)

Abbreviations: CyP: Cyclophosphamide and prednisone; MP: Melphalan and prednisone; VBCMP: vincristine, carmustine, cyclophosphamide, melphalan, prednisone; VBAD: vincristine, bleomicine, adriamycine, dexamethasone; VAD: vincristine, adriamycine, dexamethasone; PI: proteasome inhibitor; VD: bortezomib, dexamethasone; VMP: bortezomib, melphalan, dexamethasone; VCD: bortezomib, cyclophosphamide, dexamethasone; PAD: bortezomib, adriamycine, dexamethasone; IMID: immunomodulator; TD: thalidomide, dexamethasone; TCD: thalidomide, cyclophosphamide, dexamethasone; TAD: thalidomide, adriamycine, dexamethasone; Rd: lenalidomide, dexamethasone; VTD: bortezomib, thalidomide, dexamethasone; VRD: bortezomib, lenalidomide, dexamethasone; ASCT: autologous stem cell transplantation. * Age at diagnosis was unknown in 42; ** and 4 (a), 3 (b), 3 (c) and 8 patients (d) did not receive treatment due to performance status at the moment of diagnosis.

**Table 3 cancers-15-01558-t003:** Univariable analysis: long survivors vs. early death patients.

	Long Survivors (*n* = 132)	Early-Death (*n* = 252)	OR (95% CI), *p* Value
Age at diagnosis > 65 years, *n* (%)	22/132 (16.7)	163/224 (72.8)	13.4 (7.8–23.0), 0.000
Male, *n* (%)	66/132 (50.0)	136/252 (54.0)	1.2 (0.8–1.8), 0.460
IgG subtype, *n* (%)	82/129 (63.6)	122/248 (49.2)	1.8 (1.2–2.8), 0.008
IgA subtype, *n* (%)	21/129 (16.3)	167/248 (67.3)	2.5 (1.5–4.3), 0.001
Bence-Jones subtype, *n* (%)	18/129 (14.0)	36/248 (14.5)	1.0 (0.6–1.9), 0.882
Paraprotein ≥ 3 g/dL, *n* (%)	57/102 (55.9)	120/225 (53.3)	0.9 (1.6–1.4), 0.668
PCs in BM ≥ 30, *n* (%)	50/109 (45.9)	134/226 (59.3)	1.7 (1.1–2.7), 0.021
ECOG PS 2–4, *n* (%)	12/71 (16.9)	133/220 (60.5)	7.5 (3.8–14.8), 0.000
Hb ≤ 10 g/dL, *n* (%)	34/120 (28.3)	126/244 (51.6)	2.7 (1.7–4.3), 0.000
Cr ≥ 2 mg/dL, *n* (%)	10/126 (7.9)	84/244 (34.4)	6.1 (3.0–12.2), 0.000
Ca ≥ 11 mg/dL, *n* (%)	12/108 (11.1)	63/240 (26.3)	2.8 (1.5–5.5), 0.002
Lytic lesions, *n* (%)	82/125 (65.6)	151/223 (67.7)	1.1 (0.7–1.8), 0.688
Albumine < 3.5 g/dL, *n* (%)	37/112 (33.0)	125/240 (52.1)	2.2 (1.4–3.5), 0.001
β_2_ microglobulin ≥ 3.5 mg/dL, *n* (%)	28/98 (28.6)	144/190 (75.8)	7.8 (4.5–13.7), 0.000
β_2_ microglobulin ≥ 5 mg/dL, *n* (%)	11/98 (11.2)	106/190 (55.8)	10.0 (5.0–19.9), 0.000
Elevated LDH, *n* (%)	27/64 (42.2)	76/136 (55.9)	1.7 (1.0–3.2), 0.072
High-risk cytogenetic *, *n* (%)	11/97 (11.3)	30/107 (28.0)	3.0 (1.4–6.5), 0.004
ISS-1, *n* (%)	55/106 (51.9)	33/200 (16.5)	5.5 (3.2–9.3), 0.000
ISS-2. *n* (%)	40/106 (37.7)	56/200 (28.0)	0.6 (0.4–1.1), 0.082
ISS-3, *n* (%)	11/106 (10.4)	111/200 (55.5)	10.8 (5.4–21.3), 0.000

Abbreviations: OR: odds ratio; CI: confidence interval; ECOG: Eastern Cooperative Oncology Group; PCs: plasma cells; BM: bone marrow; PS: performance status; Hb: hemoglobin; Hb: hemoglobin; Cr: creatinine; Ca: calcium; LDH: lactate dehydrogenase; ISS: International Staging System. * High-risk cytogenetic included t(4;14), t(14;16) and del17p.

**Table 4 cancers-15-01558-t004:** Multivariable analysis: long survivors vs. early death.

	Long Survivors (*n* = 43)	Early Death (*n* = 82)	OR (95% CI), *p* Value
Age at diagnosis > 65 years, *n* (%)	29/43 (67.4)	19/82 (23.2)	12.2 (3.6–41.5), 0.000
Male, *n* (%)	-	-	-
IgG subtype, *n* (%)	-	-	-
IgA subtype, *n* (%)	5/82 (11.6)	26/82 (31.7)	5.3 (1.2–23.4), 0.028
Bence-Jones subtype, *n* (%)	-	-	-
Paraprotein ≥ 3 g/dL, *n* (%)	-	-	-
PCs in BM ≥ 30, *n* (%)	22/43 (51.2)	52/82 (63.2)	1.1 (0.3–3.3); 0.904
ECOG PS 2–4, *n* (%)	7/43 (16.3)	39/82 (47.6)	4.0 (1.1–14.2), 0.030
Hb ≤ 10 g/dL, *n* (%)	11/43 (25.6)	41/82 (50.0)	1.5 (0.5–4.8), 0.492
Cr ≥ 2 mg/dL, *n* (%)	2/43 (4.7)	23/82 (28.0)	3.5 (0.6–21.3), 0.170
Ca ≥ 11 mg/dL, *n* (%)	2/43 (4.7)	18/82 (22.0)	7.1 (1.2–23.4), 0.074
Lytic lesions, *n* (%)	-	-	-
Albumine < 3.5 g/dL, *n* (%)	-	-	-
β_2_ microglobulin ≥ 3.5 mg/dL, *n* (%)	-	-	-
β_2_ microglobulin ≥ 5 mg/dL, *n* (%)	-	-	-
Elevated LDH, *n* (%)	-	-	-
High-risk cytogenetic *, *n* (%)	5/43 (11.6)	25/82 (30.5)	6.1 (1.2–31.0), 0.028
ISS-1, *n* (%)	24/43 (55.8)	8/82 (9.8)	4.8 (1.4–16.6), 0.012
ISS-2. *n* (%)	-	-	-
ISS-3, *n* (%)	-	-	-

Abbreviations: OR: odds ratio; CI: confidence interval; ECOG: Eastern Cooperative Oncology Group; PCs: plasma cells; BM: bone marrow; PS: performance status; Hb: hemoglobin; Hb: hemoglobin; Cr: creatinine; Ca: calcium; LDH: lactate dehydrogenase; ISS: International Staging System. * High-risk cytogenetic included t(4;14), t(14;16) and del17p.

## Data Availability

Due to the sensitive nature of the data, information created during and/or analyzed during the current study is available from the first and corresponding authors.

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
