# Peer review of "Novel Agents as Main Drivers for Continued Improvement in Survival in Multiple Myeloma"

_cancers, 2023, doi:10.3390/cancers15051558_

Round 1
Reviewer 1 Report
Review report
The improved prognosis of MM has been realized in the daily clinical practice, so it is important to present the data in this way.
Although the theme itself is not very new, and many similar retrospective analysis studies have already been published, the detailed follow-up data over 40 years including very recent data and the large number of patients with a limitation as single center study make it well worth publishing as real-world data. In addition, this study is very useful because it visually presents a marked improvement in prognosis.
However, there are several problems with this paper. The authors need to answer these and revise the manuscript including the answers.
The authors report a worse prognosis for patients over 70 years of age, but do not show the reason why the age cutoff was set at 70 years. In routine clinical practice, ASCT is often performed below 70 years of age, but 65 years is the usual cutoff for ASCT in clinical trials. Therefore, the authors should provide a clear reason for using 70 years as the cutoff in Materials and Methods.
The authors used 65 years as the cutoff for univariate analysis comparing long survivor and early death, even though they used 70 years as the cutoff. Please explain this discrepancy.
Why is the median OS of early death 2 years in Results, even though Materials and Methods defines early death as death within 2 years? I would think that median would be shorter than 2 years.
The authors mention that prognosis are different depending on response, can you add this data to Table 2?
Why did‘t the authors use revised ISS other than ISS although FISH and LDH data are available?
Why is the number of patients different in univariate analysis in Table 2 and multivariate analysis in Table 3?
The use of novel agents have certainly improved OS, but the survival curves within the first 24 months of treatment in the 70+ cohort do not diverge much and appear to almost superimpose. There seems to be an unmet medical need here, and I would like to see a discussion on this point.
According to Table 2, the main factors associated with early death seem to be high tumor burden and poor general condition represented by lower PS which might be due to high tumor burden. Please discuss whether this is due to rapid tumor growth or late detection of the disease.
Author Response
- The authors report a worse prognosis for patients over 70 years of age, but do not show the reason why the age cutoff was set at 70 years. In routine clinical practice, ASCT is often performed below 70 years of age, but 65 years is the usual cutoff for ASCT in clinical trials. Therefore, the authors should provide a clear reason for using 70 years as the cutoff in Materials and Methods.
Thanks for the suggestion. There is not an exact cutoff age for ASCT but in the past when no new drugs were available, our policy was to offer ASCT to patients up to 70 years if there is not any comorbidity. This is the reason why we use this age as cutoff. We have added this information to the Methods section.
- The authors used 65 years as the cutoff for univariate analysis comparing long survivor and early death, even though they used 70 years as the cutoff. Please explain this discrepancy.
We appreciate the comment. In this analysis, we wanted to evaluate survival and early death but not linked to ASCT and this is the reason why we select 65 years.
- Why is the median OS of early death 2 years in Results, even though Materials and Methods defines early death as death within 2 years? I would think that median would be shorter than 2 years.
Thanks for the correction. The reviewer is right, the median OS is shorter than 2 years. We have removed the word median in the Results section.
- The authors mention that prognosis is different depending on response, can you add this data to Table 2?
Thanks for the suggestion. I think that the reviewer write table 2 referring to tables 3 and 4. Table 2 showed the treatment data. The univariate and multivariate analysis (tables 3 and 4) was performed to try to identify baseline characteristics of the patients associated with likelihood of being long survivor (static criteria) at the moment of diagnosis. However, in our opinion, response is a dynamic criterion, and therefore, we would not prefer include this variable in the analysis.
- Why did the authors not use revised ISS other than ISS although FISH and LDH data are available?
We have not used the R-ISS due to the loss of biological data, especially the LDH not available in all patients. According to the R-ISS3 we would have classified fewer than 600 patients, therefore, we would have lost many patients, especially in the first two decades (1980-1990 and 1991-2000).
- Why is the number of patients different in univariate analysis in Table 2 and multivariate analysis in Table 3?
In the multivariate analysis, we only included patients with all the baseline characteristics that were significant in the univariate analysis.
- The use of novel agents has certainly improved OS, but the survival curves within the first 24 months of treatment in the 70+ cohort do not diverge much and appear to almost superimpose. There seems to be an unmet medical need here, and I would like to see a discussion on this point.
Thanks for the comment. During the first months after diagnosis of the disease, a significant number of patients die of disease-related complications. This is more frequent in patients over 70 years of age due to their frailty. Recommended treatment dose are not always well-tolerated in this population, leading to discontinuation and worse control of the disease. To cover this issue, the use of frailty scores in daily practice and to adapt the treatment to the frailty are essentials. In addition, the snowball approach would be a useful strategy in frail patients. This approach consists of adding drugs progressively as the patient’s performance status improves until the optimal treatment is achieved. We have added this reflection to the discussion.
- According to Table 2, the main factors associated with early death seem to be high tumor burden and poor general condition represented by lower PS which might be due to high tumor burden. Please discuss whether this is due to rapid tumor growth or late detection of the disease.
Thanks for the comment. As is shown in the Table 1, patients diagnosed in the early decades more frequently presented anemia, hypercalcemia, renal failure and a poorer performance status than those diagnosed more recently. This was probably due to late detection of the disease. In contrast, in the most recent decades, the higher tumor burden was mainly due to the aggressiveness of the disease. Of note, the worse performance status at diagnosis was more strongly associated with patients older than 70 years.
Reviewer 2 Report
In the present study, Authors show the continuing increase in survival of MM patients during last decades regardless of the age of diagnosis, although improvements are more evident among youngers. In particular, this retrospective study highlights the role of novel agents in the frontline setting as main responsible for survival improvements in MM patients thus further supporting the use of most effective combinations in first-line therapy, as established. Overall this analysis on large cohort of MM patient's with a long follow up provides evidence and shows the evolution of myeloma patients over time in a real-world setting. Nevertheless, findings should be more addressed to fully support Author’s conclusion.
The bigger concern refers to OS analyses performed on different “geological eras” in MM treatment (before and after 2000). Indeed, the availability of multiple therapeutic options after progression and crossover to receive investigational agents make these analyses quite confounding and inappropriate. According to the natural history of MM, almost all patients benefit of several anti-MM drugs, including novel approaches. As results, it is opinion of this referee that OS analysis is not the appropriate end point for this study but rather the PFS or progression-free survival 2 (PFS-2), defined as the time from diagnosis to progression on first subsequent therapy: the heterogeneous therapeutic scenario significantly affects conclusions on upfront therapies
Minor points are:
· A response rate table summarizing each treatment would be beneficial
· In the table 1 on baseline characteristics of the entire cohort misses Bence-Jones subtypes, albumin and β2microglobulin levels descriptions
· Please check typos mistakes and text spacing throughout the manuscript
Author Response
- The bigger concern refers to OS analyses performed on different “geological eras” in MM treatment (before and after 2000). Indeed, the availability of multiple therapeutic options after progression and crossover to receive investigational agents make these analyses quite confounding and inappropriate. According to the natural history of MM, almost all patients benefit of several anti-MM drugs, including novel approaches. As results, it is opinion of this referee that OS analysis is not the appropriate end point for this study but rather the PFS or progression-free survival 2 (PFS-2), defined as the time from diagnosis to progression on first subsequent therapy: the heterogeneous therapeutic scenario significantly affects conclusions on upfront therapies.
We appreciate the comment. We agree with the reviewer's reflection. Survival of patients in recent decades is affected by the introduction of novel approaches that are more effective and better tolerated, both first-line and relapsed. However, we wanted to visualize the need to treat patients with the best first-line option as reported by Fonseca et al (10.1186/s12885-020-07503-y) and Yong et al (DOI: 10.1111/bjh.14213). We decided to use OS instead PFS or PFS-2 to get an overview of how OS has improved in patients with multiple myeloma reflecting also how in the most recent eras patients are able to receive more lines of therapy than in the past because of new approaches and this translates in the OS improvement. The use of PFS or even PFS 2 would restrict the analysis to the efficacy of the first and second lines of therapy.
- A response rate table summarizing each treatment would be beneficial
Thanks for the suggestion. We have included the following table to the supplementary data.
Supplementary table 1. Response after induction according to the treatment divided by age at diagnosis.
|
Age ≤70 years |
Age >70 years |
||
≥PR after induction (%) |
≥CR after induction (%) |
≥PR after induction (%) |
≥CR after induction (%) |
|
Conventional therapies |
80.3 |
20.1 |
68.0 |
16.8 |
Novel agents |
92.8 |
39.6 |
87.1 |
34.7 |
OR [95% CI], P value |
3.1 [1.9-5.2], <0.001 |
2.6 [1.8-3.7], <0.001 |
3.2 [1.7-6.1], <0.001 |
2.6 [1.4-4.8], 0.002 |
Single novel agent inductions |
85.7 |
28.6 |
80.7 |
26.5 |
At least two novel agent inductions |
96.9 |
46.0 |
100.0 |
51.2 |
OR [95% CI], P value |
5.2 [2.1-12.8], <0.001 |
2.1 [1.3-3.4], 0.001 |
Not estimated, 0.001 |
2.9 [1.3-6.4], 0.007 |
Abbreviations: CI: confidence interval; CR: complete response; OR: odds ratio; PR: partial response.
- In the table 1 on baseline characteristics of the entire cohort misses Bence-Jones subtypes, albumin and β2microglobulin levels descriptions
We appreciate the suggestion. MM BJ subtype is referred in the table 1 as a light chain only MM. We could infer albumin and β2microglobulin levels according to the ISS. However, following reviewer’s recommendation, we have added this information to table 1. ANOVA test was used to compare median of quantitative variables.
Albumin, g/dL, mean (SD) |
3.6 (±0.7) |
3.6 (±0.7) |
3.7 (±0.7) |
3.5 (±0.7) |
3.6 (±0.7) |
0.476 |
β2 microglobulin, mg/dL, mean (SD) |
5.8 (±5.4) |
5.5 (±4.1) |
6.6 (±7.6) |
4.7 (±4.2) |
6.5 (±5.2) |
0.000 |
- Please check typos mistakes and text spacing throughout the manuscript
Thanks for the comment. These mistakes have been corrected in the uploaded manuscript.
Reviewer 3 Report
Enjoyed reading the interesting report of several decades of historical data.
In the discussion form line 280 the authors quoted other similar studies, but after only one sentence start to repeat the results. I think a review and a more detailed comparisons with these published datasets would be of more value.
I think there is some statistical interaction between age, performance and phisician choice of single or double novel agent induction, I suggest a bit of elaboration about this issue, because this can affact the outstanding results with the combination.
Personally I would be interested in PI vs Imid single agent induction results and not merging the two, as these are very different treatment with very different target populations.
Other comments:
line 24 sound funny: „that favorably and independently affected”
line 159 that was only a trend, cannot state that it was longer
line 168 "therapeutic abscence", suggest "did not receive any treatment due to poor performance state"
line 256 suggest "over 10 years"
line 309-10 is confusing
were BM PC assessed by biops always or some patients had aspiration?
Author Response
- In the discussion form line 280 the authors quoted other similar studies, but after only one sentence start to repeat the results. I think a review and a more detailed comparisons with these published datasets would be of more value.
Thanks for the suggestion. A comparison with other datasets has been added in the discussion section.
- I think there is some statistical interaction between age, performance and phisician choice of single or double novel agent induction, I suggest a bit of elaboration about this issue, because this can affect the outstanding results with the combination.
We appreciate the suggestion. The reviewer is right. Novel agents have been underused in patients older than 70 years, especially during the period 2001-2010. The reasons were inexperience in the management with these drugs and lack of evidence of effectiveness and tolerability in elderly patients, as they are underrepresented in clinical trials. Many patients older than 70 years received at least two novel agents in the last decade (2011-2020), especially since the introduction of anti-CD38 monoclonal antibodies in the first line of therapy.
- Personally I would be interested in PI vs IMID single agent induction results and not merging the two, as these are very different treatment with very different target populations.
Thanks for your comment and although we recognize it could be interested to evaluate the efficacy of PIs or IMiD’s as single agents, as it is mentioned in Material and Methods section, we decided to group them in order to make possible to reach our primary objective that it was overall survival.
- line 24 sound funny: “that favorably and independently affected”
We have removed the word independently.
- line 159 that was only a trend, cannot state that it was longer
Thanks for the suggestion. The sentence has been changed and now it reads as:
“Also, the median OS of patients diagnosed in 2011-2020 tended to be longer than that of those diagnosed during 2001-2010 (28.3 months; HR 1.3 [95% CI, 1.0-1.8]; P=0.060)”
- line 168 "therapeutic abscence", suggest "did not receive any treatment due to poor performance state"
We appreciate the suggestion. We have changed the sentence.
- line 256 suggest "over 10 years"
Thanks for the comment. The word “over” has been added.
- line 309-10 is confusing
We appreciate the suggestion. These lines have been re-written and now reads as follows:
“In this regard, almost 100% and 50% of patients treated with the combination of at least two novel agents in induction achieved at least PR and CR, respectively, and presented excellent OS”
- were BM PC assessed by biops always or some patients had aspiration?
We used bone marrow aspirate for the diagnosis of multiple myeloma. We only performed bone marrow biopsy when patients presented clinical and analytical signs suggestive of myeloma and <10% infiltration in the bone marrow smear.